# Content and Uptake of Ash and Selected Nutrients (K, Ca, S) with Biomass of *Miscanthus × giganteus* Depending on Nitrogen Fertilization

Izabela Gołąb-Bogacz [1], Waldemar Helios [2], Andrzej Kotecki [2], Marcin Kozak [2] and Anna Jama-Rodzeńska [2,*]

[1] Bugaj Sp. z o.o, Bugaj Zakrzewski 5, 97-512 Kodrąb, Poland; iza.golab@o2.pl
[2] Institute of Agroecology and Plant Production, Wroclaw University of Environmental and Life Sciences, Pl. Grunwaldzki 24A, 50-363 Wrocław, Poland; waldemar.helios@upwr.edu.pl (W.H.); andrzej.kotecki@upwr.edu.pl (A.K.); marcin.kozak@upwr.edu.pl (M.K.)
* Correspondence: anna.jama@upwr.edu.pl; Tel.: +48-320-1627

**Abstract:** Fertilisation has a significant impact not only on the yielding, but also on the quality of the harvested biomass. Among energy crops, *Miscanthus × giganteus* are some of the most important plants used for combustion process. The chemical composition of biomass has significant impact on the quality of combustion biomass. The effect of nitrogen fertilisation (with dose of 60 kg N ha$^{-1}$) in different terms of biomass sampling on the content and uptake of crude ash, potassium, calcium and sulphur by rhizomes, stems, leaves and the aboveground part of miscanthus was evaluated in the paper. Nitrogen fertilisation contributed to the increase of ash content in the rhizomes and the aboveground part of plants. Independently of nitrogen fertilisation potassium content decreased in the whole vegetation period; in the case of stems this decrease amounted 60%. Calcium content in various parts of plants was highly differentiated compared to potassium content. Average calcium content in the aboveground parts was 2.68 higher compared to rhizomes. Nitrogen fertilisation affected significantly on potassium, calcium and sulphur uptake in all examined parts of plants (except stems in the case of calcium uptake). Uptake of crude ash under nitrogen fertilisation was significantly higher in all examined parts of plants during the whole vegetation period.

**Keywords:** aboveground; belowground part of *Miscanthus × giganteus*; ash; potassium; calcium; sulphur content; uptake



## 1. Introduction

The need to counteract and prevent increasingly rapid climate change is leading to the implementation of processes that will reduce greenhouse gas emissions by replacing fossil fuels with renewable energy sources. Besides the continued use of non-renewable fossil fuels, which include hard coal, lignite, natural gas and oil, energy from renewable sources is increasingly used. The acquisition of renewable energy sources is currently directed towards agriculture [1–8].

Energy from plant biomass is mainly obtained by pyrolysis, gasification or direct combustion of appropriately ground or granulated mass [9,10]. Miscanthus (*Miscanthus × giganteus* Greef et Deuter) can play a significant role as a source of renewable energy for Europe [11–13]. Obtaining high quality biomass for the combustion process depends on the quality of the raw material (biomass) [14,15], while the quality of the raw material depends on the content of various elements (for example, high lignin content is desirable for thermochemical and undesirable for biochemical processes) [16,17].

The content of elements in the biomass is significantly influenced by genetic properties [14,18] which can be modified by environmental conditions, such as soil properties, pH, weather conditions (precipitation, temperature), as well as agrotechnical treatments—mainly fertilisation [19–23]. Date of harvest (late winter or spring) can also contribute to the

reduced content of nutrients that results from their translocation from aboveground part of plant to rhizomes or natural leaching of components from leaves and stems [23–25]. Appropriate chemical composition, especially low content of contaminants in biomass, is desirable during harvest, especially for biomass for thermal combustion, as it contributes to the minimisation of their emissions [23].

Most of the available studies on the content and nutrient uptake of miscanthus concern nitrogen, phosphorus, potassium and magnesium [25–28], while only a few works concern calcium and sulphur content [29,30]. An innovative part of the study was to examine the dynamics of sulphur uptake during the whole vegetation period, taking into account nitrogen fertilisation in various parts of plants.

Crude ash content and examined macroelements have a significant impact on the quality of biomass combustion; therefore, the relevance of these elements is discussed. High ash concentration decreases the heating value [31,32]. Potassium, alongside silicon, is the main component of ash [12]. The potassium content of biomass is very important because its high content can increase the corrosion effect in heating systems and lower the melting point of ash [31], and is regarded as a critical element in ash-related problems [32]. Therefore, the potassium content should be as low as possible [32]. For optimal plant growth, the potassium content should be 10–50 g of DM [31]. Sulphur also plays an important role during the combustion process. Sulphur compounds that are formed during this process lead to corrosion and are emitted into the atmosphere [30]. In turn, calcium can inhibit the occurrence of silicate melt-induced slagging and bed agglomeration, as a result of forming melting calcium potassium phosphates and silicates at high temperatures [30–32].

The work hypothesis assumes that fertilisation in a of dose 60 kg ha$^{-1}$ will contribute to changes in content an uptake of selected macronutrients and ash. It has been estimated that particular parts of the plant (rhizomes, stems, leaves) will be characterised by different ash, Ca, K, and S accumulation. Additionally, fertilisation at a dose 60 kg ha$^{-1}$ N causes the increase in uptake of ash and selected macroelement.

The aim of the study was to determine the effect of nitrogen fertilisation on the content and uptake of ash and selected macroelements in *Miscanthus × giganteus*.

## 2. Materials and Methods

### 2.1. Study Site and Materials

The experiment with miscanthus and nitrogen fertilisation started by separating plots on the plantation carried out in 2004. Detailed information is contained in the article by Bogacz et al. 2020 [33]. The study with miscanthus was conducted in the years 2014–2016 at Experimental Station belonging to Wroclaw University of Environmental and Life Sciences, Pawlowice (geographical location 17°7′ E and 51°08′ N in the Lower Silesian Voivodship (Figure 1)). The tested factor was nitrogen fertilisation (0, 60 kg ha$^{-1}$ N). Miscanthus sampling started from the 30th day of the vegetation period and was done every 30 days until the end of the vegetation period (June, July, August, September, October, November and December). At each date of sampling, a plant sample of the aboveground part of the plant and rhizomes was sampled from an area of 0.25 m$^2$. Samples for chemical analysis were reduced according to the standard requirements of PN-EN 96 ISO 14780:2017-07 [34] (which defines methods for reducing combined samples to laboratory samples and laboratory samples to sub-samples and general analysis samples, and is applicable to solid biofuels). Plant samples were sampled from the area of 0.25 m$^2$ by gentle extraction of rhizomes from the soil with the whole stems. Dry mass for laboratory samples was determined by air-drying the dry mass at 105 °C for three hours according to Polish standard (PN-R-04013:1988).

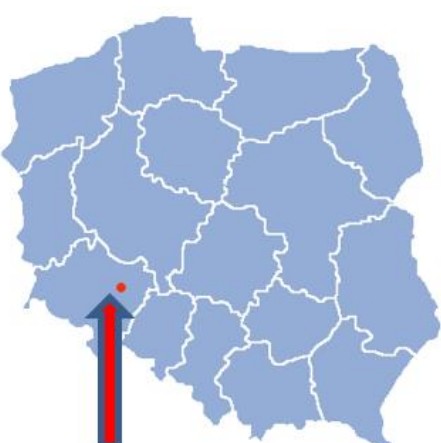
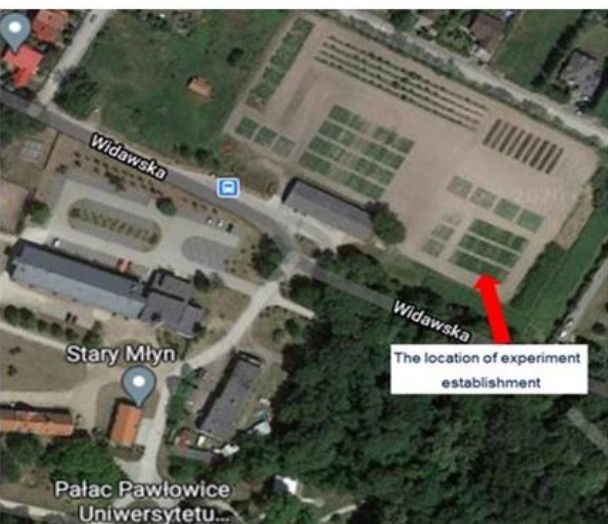

**Figure 1.** Location of experiment.

The weather and soil condition, experiment design and agrotechnical treatments are described in research by Bogacz et al. [33].

### 2.2. Chemical Analysis of Plant Material

The content of ash and macroelements in plant material was determined in the laboratory belonging to the Institute of Agroecology and Plant Production. The content of crude ash and macroelements in the aboveground part was calculated on the basis of the content of these elements in the leaves and stems, taking into account the structure of the dry matter yield.

Chemical analyses comprised:

- crude ash by burning dry plant material at 600 °C in an electric furnace: incineration of plant material and combustion 1/2 g weighing the analytical sample of plant material in the muffle furnace at 600 ± 15 °C and baking the remaining ash;
- potassium and calcium on the flame photometer (BWB Technologies UK LTD), using flame photometry; mineralization of plant material through the use of sulphuric acid and perhydrol and subsequent determination on a flame photometer;
- total sulphur by nephelometric method, after wet mineralisation with concentrated sulphuric acid with 30% perhydrol, by the Bradley–Lancaster nephelometric method.

Uptake of crude ash and selected elements vas calculated based on yield biomass and chemical content of the examined parts of plants.

### 2.3. Statistical Analysis

The experiment was conducted in four replications in order to test the effects of N fertilisation on the content and uptake of ash and macroelements in *Mischanthus giganteus*. The analysis of variance (ANOVA) and the mixed model with repeated measurements were used. Doses of nitrogen fertilisers were assumed to be a fixed factor, while years was assumed to be random. The results of chemical analysis of the Mischanthus were analysed by ANOVA in the Statistica program (13.1 StatSoft, Kraków, Poland). One-way ANOVA (nitrogen fertilisation, then year of experiment) was performed including post-hoc analysis. The level of significance was determined as $p < 0.05$.

Homogeneous groups were determined on the basis of the Tukey test. The groups were determined from the lowest to the highest value. The correlation of repeated measurements was performed as the average value over the three-year growing season of each month. The *p*-value concerns the subsequent months.

### 3. Results

#### 3.1. Crude Ash Content and Uptake

The effect of nitrogen fertilisation on ash content in the rhizomes ($p = 0.0035$), stems ($p = 0.0002$) and aboveground part of *Miscanthus × giganteus* ($p < 0.001$) except for the leaves was found. Even though rhizomes are not involved in the combustion process, knowledge of the ash content of rhizomes allowed the ash content to significantly increase from 2014 to 2016 in rhizomes ($p = 0.0156$), whereas the highest content was found in the leaves ($p = 0.0312$) in 2015 (the lowest annual sum of precipitation—392 mm). The highest content of ash was observed in the aboveground part of plants in the first year ($p = 0.0047$). The highest content of this component was found in leaves, which is particularly beneficial as the stem has the greatest share in the process of biomass combustion (Table 1). The highest content of crude ash was found at the beginning of the vegetation period, and as the plants developed (and also as a result of the ageing processes), its content decreased. The decrease in ash content in stems was greater than in leaves at the beginning of the vegetation period (Figure 2). The figures show the significance values of differences (*p*-values) of ash content in subsequent months of observation for control and dose 60 (Figure 2).

**Table 1.** Crude ash content in dry matter of miscanthus in g kg$^{-1}$ (average for the years 2014–2016).

| Dose kg ha$^{-1}$ N | Rhizomes | Stems | Leaves | Aboveground Part |
|---|---|---|---|---|
| 0 | 43.6 a | 37.6 a | 56.7 a | 53.6 a |
| 60 | 46.5 a | 42.2 a | 58.3 a | 57.6 a |
| *p*-value | 0.0035 | 0.0002 | 0.2418 | <0.001 |
| 2014 | 43.3 a | 39.7 a | 58.6 a | 57.3 a |
| 2015 | 45.2 a | 39.0 a | 59.3 a | 54.5 a |
| 2016 | 46.7 a | 41.0 a | 54.7 a | 55.1 a |
| *p*-value | 0.0156 | 0.1980 | 0.0312 | 0.0047 |

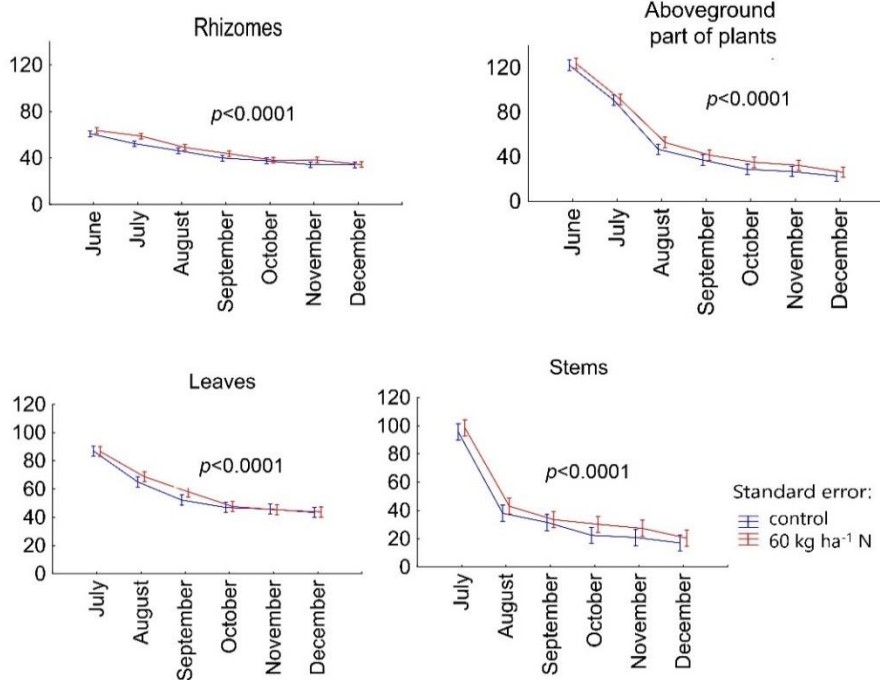

**Figure 2.** Crude ash content in examined part of miscanthus (g kg$^{-1}$) (three-year average content from measurements during the growing season every 30 days).

The crude ash uptake through individual elements of the plant was significantly dependent on the nitrogen fertilisation ($p \leq 0.001$). The highest uptake in the rhizomes ($p < 0.001$) was found in the third year, whereas the highest uptake in the stems ($p \leq 0.001$) and aboveground part of plants ($p = 0.0467$) was found in the second year of the experiment (Table 2). Crude ash accumulation by *Miscanthus × giganteus* per 1 m$^2$ in rhizomes increased throughout the entire vegetation period, while in stems and aboveground parts of the plant, it decreased at the end of the vegetation period. Nitrogen fertilization caused greater uptake of crude ash in all examined parts of plants during the whole vegetation period (Figure 3). The *p*-values presented on the figure concern the date of plant material sampling. The figures show the significance values of differences (*p*-values) of ash uptake in subsequent months of observation for control and dose 60 (Figure 3).

**Table 2.** Crude ash uptake by g·m$^{-2}$ (average for 2014–2016).

| Dose kg ha$^{-1}$ N | Rhizomes | Aboveground Part | | | Rhizomes and Aboveground Part |
| --- | --- | --- | --- | --- | --- |
| | | Stems | Leaves | All Together | |
| 0 | 44.8 a | 46.0 a | 33.9 a | 74.4 a | 119.2 a |
| 60 | 54.3 b | 60.1 b | 46.3 b | 99.0 b | 153.3 b |
| *p*-value | <0.001 | <0.001 | <0.001 | <0.001 | <0.001 |
| 2014 | 48.5 a | 49.8 a | 40.8 a | 85.5 a | 134.0 a |
| 2015 | 47.7 b | 57.9 a | 39.1 a | 89.1 a | 136.8 a |
| 2016 | 52.6 b | 51.4 a | 40.3 a | 85.5 a | 138.1 a |
| *p*-value | <0.001 | <0.001 | 0.3064 | 0.0467 | 0.1679 |

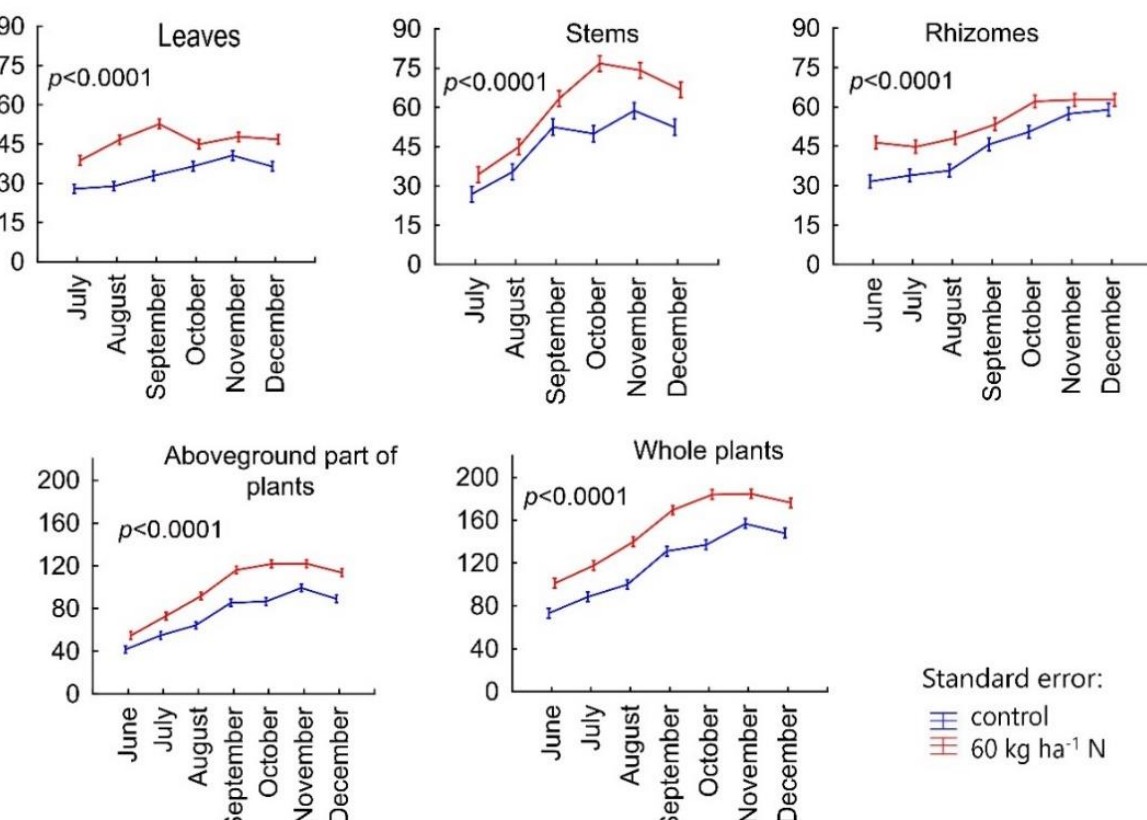

**Figure 3.** Crude ash uptake in examined part of miscanthus (g m$^{-2}$) (three-year average content from measurements during the growing season every 30 days).

### 3.2. Potassium Content and Uptake

The potassium content in leaves ($p$ = 0.0085) was significantly dependent on the nitrogen fertilisation. In the stem of *Miscanthus × giganteus*, the highest content of potassium was found in the third year of the study ($p$ = 0.0032), and in the second year in rhizomes ($p$ = 0.0219) and leaves ($p$ < 0.001) (Table 3).

**Table 3.** Potassium content in dry matter of miscanthus g kg$^{-1}$ (average for the years 2014–2016).

| Dose kg ha$^{-1}$ N | Rhizomes | Stems | Leaves | Aboveground Part |
|---|---|---|---|---|
| 0 | 12.7 a | 11.6 a | 12.3 a | 12.0 a |
| 60 | 11.9 a | 11.6 a | 13.9 a | 12.6 a |
| *p* value | 0.1455 | 0.9491 | 0.0085 | 0.1643 |
| 2014 | 12.7 a | 11.0 a | 13.6 a | 12.1 a |
| 2015 | 13.0 a | 10.3 a | 15.1 ab | 12.1 a |
| 2016 | 11.1 a | 13.5 a | 10.6 a | 12.7 a |
| *p* value | 0.0219 | 0.0032 | <0.001 | 0.4601 |

A decrease was observed in potassium content in the leaves, stems and aboveground part of *Miscanthus × giganteus* since August to the December. The lowest level of this element was found in December, when the potassium content in the aerial part of plants was on average about twice as low as in June. In turn, a decrease in potassium content in the rhizomes was found from the beginning of vegetation period until November. The increase in potassium content in the rhizomes from November to the end of the vegetation period (Figure 4) might be the result of translocation of this element from the aboveground part of plants to the rhizomes. The figures show the significance values of differences ($p$-values) of potassium content in subsequent months of observation for control and dose 60 (Figure 4).

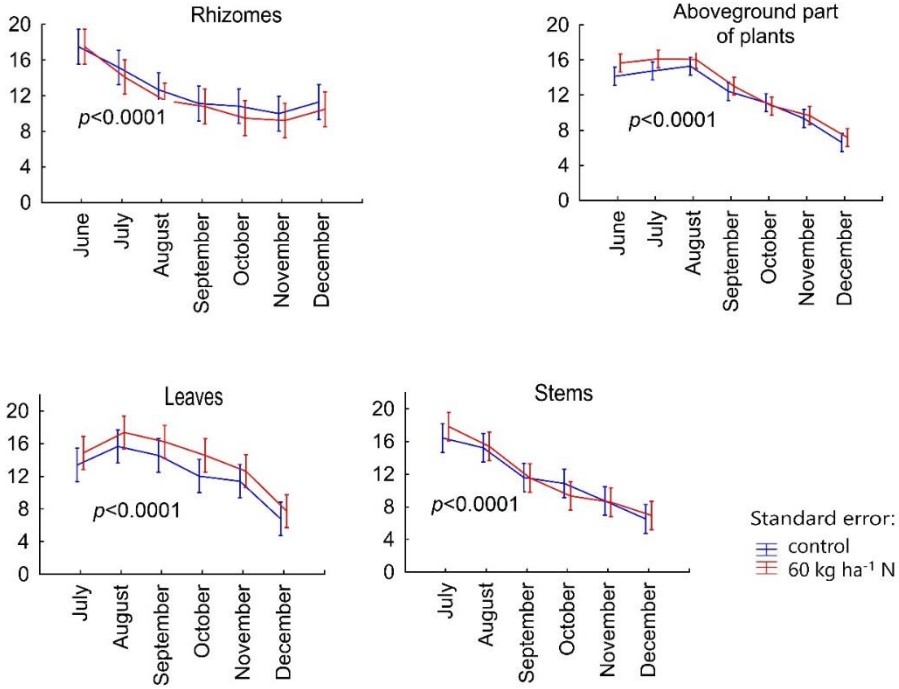

**Figure 4.** Potassium content in examined part of miscanthus (g kg$^{-1}$).

Potassium uptake (g m$^{-2}$) by *Miscanthus × giganteus* was dependent on nitrogen fertilisation and the years of the experiment. Nitrogen fertilisation caused an increase in potassium accumulation (g m$^{-2}$) in all examined parts of plants. The highest potassium

uptake was found in the rhizomes ($p < 0.001$) and the aboveground part of plants ($p < 0.001$) in the first year of research (Table 4).

**Table 4.** Potassium uptake of the giant miscanthus in g m$^{-2}$ (average for 2014–2016).

| Dose kg ha$^{-1}$ N | Rhizomes | Aboveground Parts | | | Rhizomes and Aboveground Part |
| --- | --- | --- | --- | --- | --- |
| | | Stem | Leaves | Together | |
| 0 | 13.3 a | 17.9 a | 7.6 a | 22.5 a | 35.8 a |
| 60 | 13.9 a | 19.1 a | 11.4 b | 27.1 b | 41.0 a |
| *p*-value | 0.0064 | 0.0021 | <0.001 | <0.001 | <0.001 |
| 2014 | 15.1 b | 19.7 a | 10.2 b | 26.5 a | 41.6 a |
| 2015 | 14.3 b | 16.9 a | 10.4 b | 24.1 a | 38.4 a |
| 2016 | 11.4 a | 18.9 a | 7.8 a | 23.8 a | 35.1 a |
| *p*-value | <0.001 | <0.001 | <0.001 | <0.001 | <0.001 |

Rhizomes accumulated potassium until the end of vegetation (increasing trend). Without nitrogen fertilization in the aboveground part of plants, the peak potassium uptake was observed in November, whereas the highest accumulation was seen earlier on the plots with nitrogen fertilization (Figure 5). The figures show the significance values of differences (*p*-values) of potassium uptake in subsequent months of observation for control and dose 60 (Figure 5).

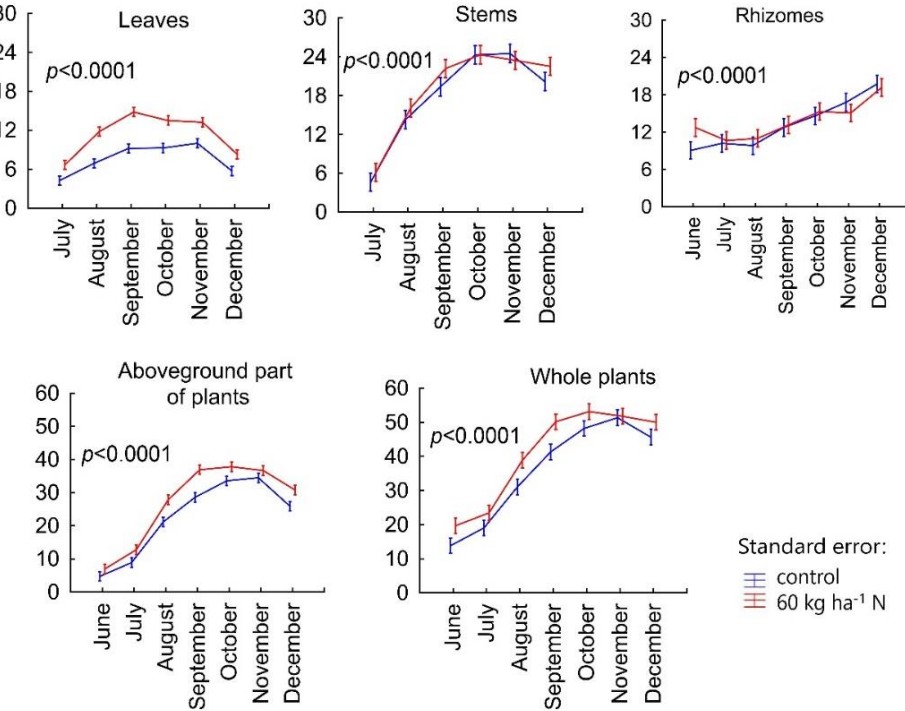

**Figure 5.** Potassium uptake in examined part of miscanthus (g m$^{-2}$) (three-year average content from measurements during the growing season every 30 days).

## 3.3. Calcium Content and Uptake

Nitrogen fertilisation had no significant effect on the calcium content in the examined parts of plants (Table 5). The year of the experiment had a significant effect on calcium content in rhizomes ($p < 0.001$), stems ($p = 0.0036$) and leaves ($p < 0.001$) (Table 5).

**Table 5.** Calcium content in dry matter of the giant miscanthus in g·kg$^{-1}$ (average for the years 2014–2016).

| Dose kg ha$^{-1}$ N | Rhizomes | Stems | Leaves | Aboveground Part |
|---|---|---|---|---|
| 0 | 0.58 a | 1.34 a | 1.78 a | 1.52 a |
| 60 | 0.55 a | 1.24 a | 1.93 a | 1.51 a |
| *p*-value | 0.5401 | 0.2250 | 0.1717 | 0.8787 |
| 2014 | 0.64 b | 1.24 a | 1.80 a | 1.51 a |
| 2015 | 0.79 b | 1.11 a | 2.31 a | 1.50 a |
| 2016 | 0.27 a | 1.52 a | 1.45 b | 1.54 a |
| *p*-value | 0.0000 | 0.0036 | <0.001 | 0.8886 |

An increase in the content of this element in rhizomes was found until August and in the stems to the end of the vegetation period (Figure 6). The figures show the significance values of differences (*p*-values) of calcium content in subsequent months of observation for control and dose 60 (Figure 6).

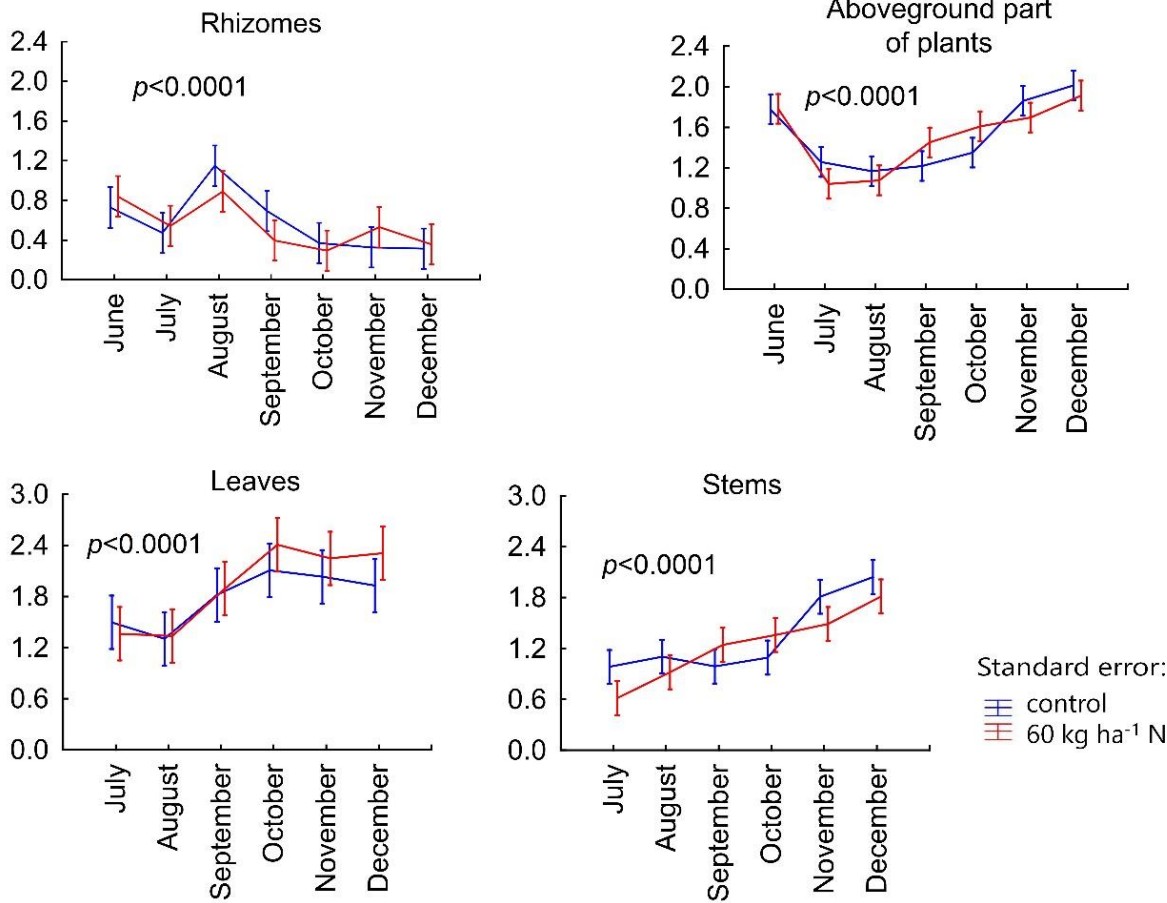

**Figure 6.** Calcium content in examined part of miscanthus (g kg$^{-1}$) (three-year average content from measurements during the growing season every 30 days).

Calcium uptake depended on nitrogen fertilisation in all parts of plants ($p < 0.001$) except stems. Significant changes in calcium uptake were found during the years of research in the rhizomes ($p \leq 0.001$), stems ($p \leq 0.001$), leaves ($p \leq 0.001$) and the whole plants ($p \leq 0.001$) (Table 6).

**Table 6.** Calcium uptake by the giant miscanthus in g·m$^{-2}$ (average for the years 2014–2016).

| Dose kg ha$^{-1}$ N | Rhizomes | Aboveground Parts | | | Rhizomes and Aboveground Parts |
|---|---|---|---|---|---|
| | | Stems | Leaves | All Together | |
| 0 | 0.56 a | 2.80 a | 1.20 a | 3.51 a | 4.07 a |
| 60 | 0.61 a | 2.80 a | 1.71 b | 3.98 a | 4.59 a |
| *p*-value | <0.001 | 0.9045 | <0.001 | <0.001 | <0.001 |
| 2014 | 0.64 b | 2.81 a | 1.44 ab | 3.76 a | 4.40 a |
| 2015 | 0.83 c | 2.41 a | 1.73 a | 3.64 a | 4.47 a |
| 2016 | 0.28 a | 3.18 a | 1.19 b | 3.84 a | 4.12 a |
| *p*-value | <0.001 | <0.001 | <0.001 | 0.0573 | <0.001 |

An increase in calcium uptake was seen in the stems and aboveground part of plants through the entire vegetation period. Changes in calcium uptake in the rhizomes were lower in this period compared to aerial parts of *Miscanthus × giganteus* (Figure 7). The figures show the significance values of differences (*p*-values) of calcium uptake in subsequent months of observation for control and dose 60 (Figure 7).

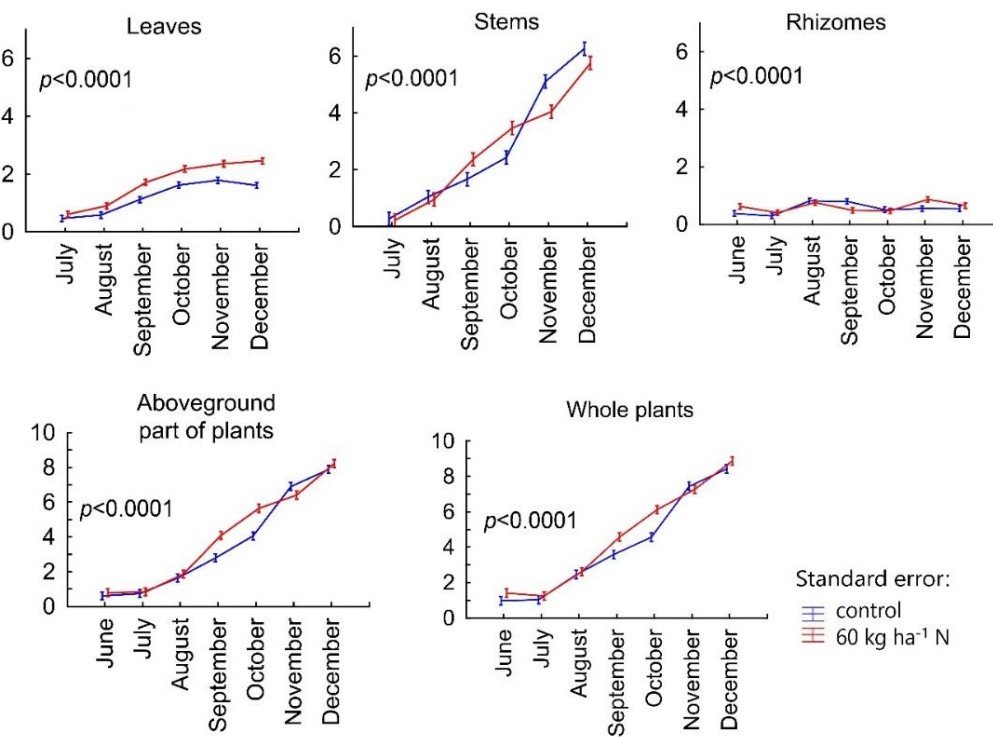

**Figure 7.** Calcium uptake in examined part of miscanthus (g m$^{-2}$) (three-year average content from measurements during the growing season every 30 days).

### 3.4. Sulphur Content and Uptake

Nitrogen fertilisation had a significant impact on sulphur content in the stems (*p* = 0.0485) and aboveground parts (*p* = 0.0067). Significant changes in sulphur content were found in the different years of the experiment for rhizomes (*p* = 0.0345), stems (*p* < 0.001), leaves (*p* < 0.001) and aboveground parts of plants (*p* = 0.0219). The highest sulphur content in the rhizomes and stems was seen in the first year of field experiments and in the leaves and aboveground part of plants in the third year (Table 7).

**Table 7.** Sulphur content in dry matter of miscanthus in g kg$^{-1}$ (average for years 2014–2016).

| Dose kg ha$^{-1}$ N | Rhizomes | Stems | Leaves | Aboveground Parts |
|---|---|---|---|---|
| 0 | 0.78 a | 0.62 a | 0.63 a | 0.69 a |
| 60 | 0.81 a | 0.67 a | 0.64 a | 0.75 a |
| *p*-value | 0.4928 | 0.0485 | 0.5357 | 0.0067 |
| 2014 | 0.90 b | 0.71 b | 0.56 a | 0.71 a |
| 2015 | 0.77 ab | 0.55 a | 0.66 ab | 0.69 a |
| 2016 | 0.72 a | 0.67 b | 0.69 b | 0.76 a |
| *p*-value | 0.0345 | <0.001 | <0.001 | 0.0219 |

The sulphur content in the aboveground parts, stems and leaves decreased with the development of plants. The dynamic changing of sulphur content in the aerial part of *Miscanthus × giganteus* was the highest at the beginning of the vegetation period. The lowest sulphur content in the rhizomes was found in October (Figure 8). The figures show the significance values of differences (*p*-values) of sulphur content in subsequent months of observation for control and dose 60 (Figure 8).

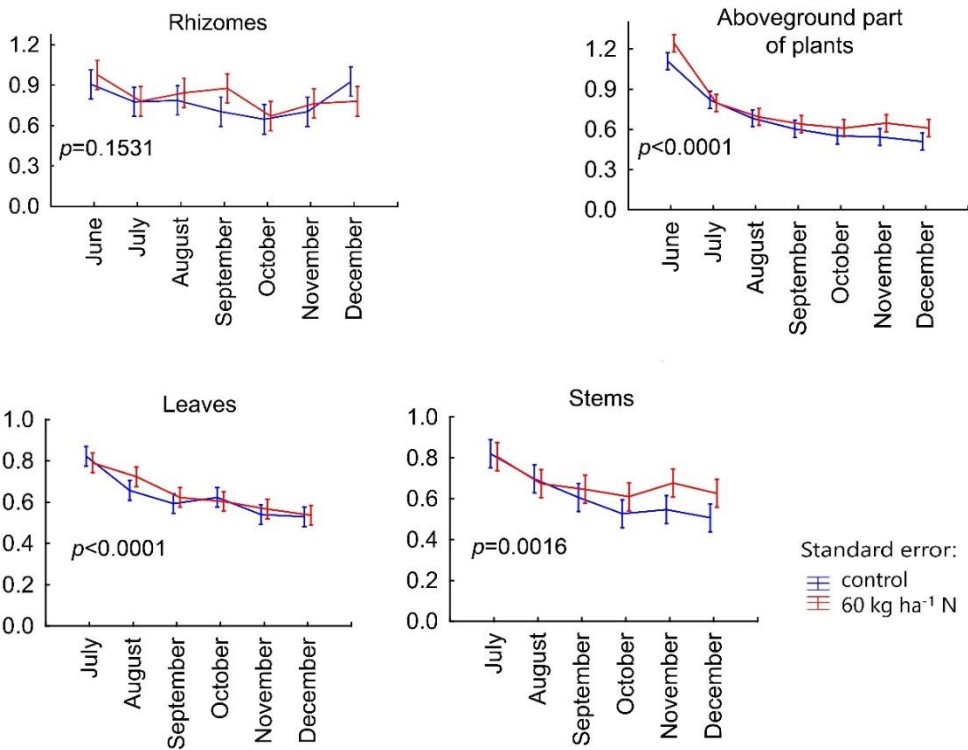

**Figure 8.** Sulphur content in examined part of miscanthus (g kg$^{-1}$) (three-year average content from measurements during the growing season every 30 days).

Sulphur uptake (g m$^{-2}$) by *Miscanthus × giganteus* was significantly dependent on nitrogen fertilisation and year of the experiment (Table 8). The highest sulphur uptake was found on plots with nitrogen fertilisation in all examined parts of plants (*p* < 0.001). The highest sulphur accumulation per m$^2$ in the rhizomes and aboveground part of plants was observed in the first year of the field experiment (Table 8).

**Table 8.** Sulphur uptake by the giant miscanthus in g·m$^{-2}$ (average for the years 2014–2016).

| Dose kg ha$^{-1}$ N | Number of Days after the Start of the Vegetation | Rhizomes | Aboveground Part | | | Rhizomes and Aboveground Parts |
| --- | --- | --- | --- | --- | --- | --- |
| | | | Stems | Leaves | All Together | |
| 0 | | 0.87 a | 1.03 a | 0.39 a | 1.27 a | 2.14 a |
| 60 | | 0.99 b | 1.28 b | 0.53 b | 1.63 b | 2.62 b |
| | *p*-value | <0.001 | <0.001 | <0.001 | <0.001 | <0.001 |
| | 2014 | 1.06 c | 1.36 b | 0.40 a | 1.57 a | 2.63 a |
| | 2015 | 0.89 ab | 0.99 a | 0.45 a | 1.30 a | 2.19 a |
| | 2016 | 0.83 a | 1.12 ab | 0.52 b | 1.48 a | 2.31 a |
| | *p*-value | <0.001 | <0.001 | <0.001 | <0.001 | <0.001 |

The highest sulphur uptake by rhizomes and stems was found in December. It should be noted that stems accumulated over 2–3 times more sulphur than leaves. Sulphur uptake in *Miscanthus × giganteus* increased with the progressing vegetation period in all parts of the field experiment (Figure 9). The figures show the significance values of differences (*p*-values) of Sulphur uptake in subsequent months of observation for control and dose 60 (Figure 9).

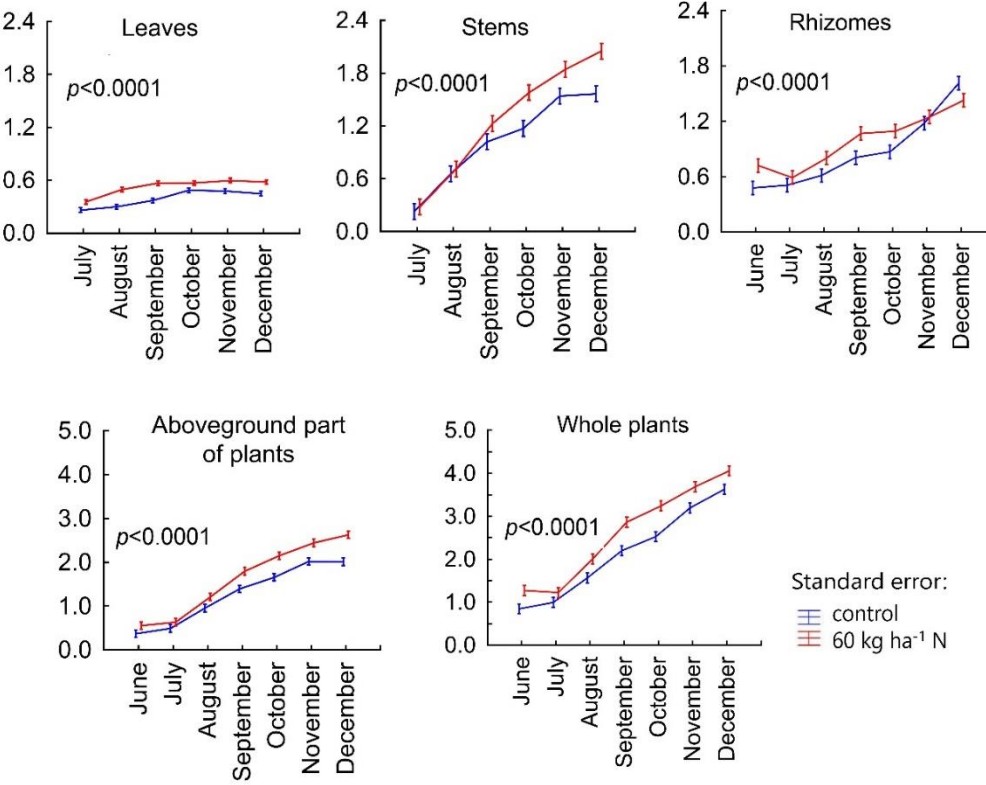

**Figure 9.** Sulphur uptake in examined part of miscanthus (g m$^{-2}$) (three-year average content from measurements during the growing season every 30 days).

## 4. Discussion

Mineral concentration plays an essential role in biomass combustion quality [14,35]. To improve biomass quality of *Miscanthus × giganteus*, cultivation practice should be based on keeping the nitrogen fertilisation rate as low as possible and delaying harvest until the

spring following growth, as this will allow nutrient remobilisation and leaching of soluble minerals like K and Cl through rainfall [35,36].

Nutrient remobilisation seems to be a good strategy for perennial rhizomatous grasses [37,38] and represents an environmentally friendly strategy to reduce fertiliser applications [36]. When calculating the nutrient balance and fertiliser recommendations, the remobilisation of nutrients within the plant must be taken into account [25]. September and March are irrelevant for nutrient remobilisation [28]. The increase in nutrient content found in rhizomes in autumn and winter may be caused by remobilisation from the aboveground parts to the underground part [25]. In our research, generally, nutrient concentrations were highest at the beginning of the growing period and decreased clearly during the growing season. There were no significant differences caused by N fertilisation (except for potassium in leaves and sulphur in stems). The large loss of K from shoots between September and harvest can be attributed to leaching from the senescent plant material as K is not organically metabolised [39]. Some leaves fell after the end of the growing period, contributing to improved properties of soil by increasing the contents of elements and organic matter, thereby leading simultaneously to a decrease in ash uptake by plants [24]. The mineral concentration of aerial biomass is at its highest during spring and early summer and then declines, probably as a result of remobilization [24]. These results are also confirmed by own research in the aboveground parts of plants. It is documented that mineral concentration in the aboveground biomass of *Miscanthus* × *giganteus* decreases gradually from autumn to winter [24,28]. Our results highlight a decline in the concentration of crude ash and macronutrients in aboveground parts of plants from spring to autumn.

The average ash content in Giant Mischanthus according to Borkowska (2007) [31] is about 27.6 g kg$^{-1}$. In the study Baxter et al. (2014) [15], the average ash content in leaves was between 40 and 60 g kg$^{-1}$ DM, and in stems the mean value was lower, between 10 and 30 g kg$^{-1}$ dm. In our research the highest average ash content was found in leaves (57.5 g kg$^{-1}$), less in rhizomes (45.1 g kg$^{-1}$) and the lowest in stems (39.9 g kg$^{-1}$). In the research by Lewandowski and Heinz (2002) [36], the content of ash in the aboveground part of plants decreased from December to February. Ash content decreased also from autumn to spring in the study of Lewandowski et al. (2003) [40]. Similarly, delayed harvesting in the research of Lewandowski and Heinz (2003) [36] contributed to a reduction in ash content by 28% on average in Portugal and Great Britain, by 42% in Germany, by 50% in Sweden and by 54% in Denmark. Kotecki et al. (2010) [41] found that nutrient and crude ash yields were higher during the autumn harvest compared to the winter harvest, rising from 31 to 69%, while nitrogen fertilisation contributed to an increase in ash content. In our research, the ash content depended on nitrogen fertilisation and years of experiment in the rhizomes and aboveground part of miscanthus. Ash content decreased during the whole vegetation period. Studies by Lewandowski and Kircherer (1997) [42] showed that miscanthus leaves have a higher ash content than stems, which is also confirmed by own research.

The content of potassium in the aboveground part of plants ranges from 4.3 to 10.5 g kg$^{-1}$ DM [22]. According to Borzęcka-Walker's (2010) [31] study, the potassium content in the aerial parts of miscanthus plants ranged from 2.7 to 9.9 g kg$^{-1}$ DM on heavy black soil and from 1.6 to 9.4 g kg$^{-1}$ DM on medium heavy black soil, depending on the genotype and year of cultivation. In the research of Kalembasa et al. (2019) [22] the mean potassium content in mischanthus grass biomass was 15.66 g kg$^{-1}$ D.M. Furthermore, Lewandowski et al. 2000 [43] presented a review of potassium content obtained in field studies by several authors for some locations in Europe. Potassium concentration was significantly influenced during harvest time. According to Jensen et al. (2017) [38], potassium content decreased over the three harvests from June (2009) to February (2010) with the highest concentration during the summer. As expected, delaying the harvest by three to four months improved the combustion quality by reducing potassium content from 9 to 4 g kg$^{-1}$ DM.

Their experiment indicated that many genotypes of *Mischanthus* are characterised by higher concentrations of potassium in autumn. According to Beale and Long (1997) [24], potassium concentrations in the aboveground dry matter decreased from 32 to 12.0 g kg$^{-1}$ during whole vegetation period. In our study, the potassium content also decreased from summer till the end of vegetation in aboveground part of plants. Kalembasa et al. (2019) [22] proved that potassium was transferred from the aboveground parts of plants to rhizomes at the end of the growing season. According to Christian et al. (2008) [44] transfer of potassium from leaves and stem to rhizomes is 14–30%.

The uptake of macronutrients is strongly dependent on the yield. The higher obtained yield, the higher the uptake of the following element [44]. In the Christian et al. (2008) [44] experiment between 1993 and 1995, the mineral uptake increased when the yield increased rapidly. The translocation of the elements during harvest depends on many external factors, especially weather conditions. While *Mischanthus* is characterised by higher dry yields (about 30 Mg ha$^{-1}$) from a three-year old crop, Beale and Long (1997) [24] found high potassium uptake gaining 38.0 g m$^{-2}$. Nassi o Di Nasso et al. (2011) [45] obtained potassium uptake of around 27.0 g m$^{-2}$. In our research potassium uptake was around 24.8 g·m$^{-2}$. Greater uptake of potassium in the Roncucci et al. (2015) [28] study was found in autumn (16.0 g·m$^{-2}$), and uptake was lower during wintertime. In this research, potassium uptake by the aboveground part of miscanthus at wintertime had values corresponding to around 33% of those recorded at autumn harvest. The time of harvest was the most relevant factor influencing miscanthus nutrient uptake in own experiments and those by Roncucci et al. (2015) [28].

Aerial parts of grasses accumulated mostly calcium, potassium and magnesium. The issue of calcium content in the rhizomes was undertaken by Stypczyńska et al. (2017) [21]. The concentration of this element in their study in the rhizomes was 1.5 g kg$^{-1}$ DM. In turn, Nassi di Nasso et al. 2010 [45] studied calcium content in the rhizomes and obtained values of 0.5–1.4 g kg$^{-1}$ DM which are confirmed in our research. The content of calcium was affected by the following factors: genotypes, geographical location of plantation and weather conditions, according to Helios (2018) [46]. In the lack of calcium fertilisation the content of this element relies on age of the plantation. In a 12-year study by Helios (2018) [46], the calcium content ranged from 3.1 g kg$^{-1}$ DM while calcium uptake amounted to 0.45 g m$^{-2}$ in the first year of experiment to 0.5 g kg$^{-1}$ DM, while the calcium uptake was 1.2 g m$^{-2}$ in the tenth year of cultivation. In studies by Baxter et al. (2014) [15] and Stypczyńska et al. (2017) [21], leaves of miscanthus were characterised by higher calcium content compared to the stems, which is confirmed by our research. In conducted experiments by Lewandowski and Kicherer [42], the calcium concentrations in the leaves ranged from 2.3 to 3.7 g kg DM while that in the stems ranged from 0.5 to 1.1 g kg DM.

In our research, the trends of changes in calcium content during vegetation were similar to those of Kotecki et al. (2010) [41] who showed that the content of this element in the aboveground part of the plant was decreasing until summer and then it increased.

Sulphur plays an important role during the combustion process. Sulphur compounds that are formed during this process lead to corrosion and are emitted into the atmosphere [30]. In Lewandowski and Kicherer's [42] research no definite effect of nitrogen fertilisation on sulphur concentration in the leaves and stems was found. In our experiment, the content of this element in the stems was dependent on examined factor ($p = 0.0485$) while the nitrogen fertilisation had no significant impact on sulphur content in the leaves. For the entire vegetation period of miscanthus, Spiak et al. (2012) [29] showed that almost half the sulphur content is present in the stems compared to that in the leaves. In the study by Baxter et al. (2014) [15], on the other hand, the opposite results were obtained. In our study, the highest sulphur content was found in the rhizomes and there was less in the leaves and stems. The sulphur content in the leaves (0.64 g kg$^{-1}$ DM) and stems (0.65 g kg$^{-1}$ DM) was similar. Concentration of this element in aboveground parts of miscanthus amounted to 0.72 g kg$^{-1}$ DM. In research by Kotecki (2010) [41], sulphur content in aerial parts of plants was 0.5–0.8 g kg$^{-1}$ DM.

In our field experiment, the highest content of this component was found in young plants. As the vegetation progressed, the sulphur content decreased in the aboveground part of plants by around 50%. In contrast to the content, the sulphur uptake was significantly higher in stems than in leaves. The uptake of sulphur in the aboveground part and whole plants with an increased trend was observed until the end of the vegetation season. A similar tendency was observed in rhizomes from July to December.

## 5. Conclusions

Because of the need to reduce emissions, and to avoid worsening the air quality by producing the compounds during combustion of, e.g., hard coal for heating purposes in many Polish cities and other Central and Eastern European countries, the low content of mineral components in *Miscanthus* × *giganteus* biomass is very desirable and may constitute an alternative source of biomass for energy purposes.

While the research hypothesis was verified, it should be stated that only ash content in rhizomes and aboveground part of plants depended significantly on nitrogen fertilisation, while potassium (except in leaves), calcium and sulphur content (except in stems and aboveground parts) were not significantly influenced by this factor. The uptake of the studied elements was significantly dependent on nitrogen fertilisation in the case of ash, potassium, sulphur and calcium (except for stems). K and S concentrations were highest at the beginning of the growing period and decreased clearly during the growing season.

The ash content was significantly higher under the influence of nitrogen fertilisation in leaves at 58.3 g m$^{-2}$ and the lowest in stems at 42.2 g m$^{-2}$, and the highest intake by stems at 60.1 g m$^{-2}$ and the lowest in leaves at 46.3 g m$^{-2}$. Significantly higher sulphur uptake was found in stems under the influence of nitrogen fertilisation at the amount of 1.28 g m$^{-2}$.

**Author Contributions:** Conceptualization: A.J.-R., I.G.-B., W.H.; Methodology: A.K., M.K., W.H.; Software: I.G.-B., W.H., A.J.-R.; Validation: A.K., M.K.; Investigation: I.G.-B., W.H., A.K.; Formal Analysis: I.G.-B., W.H., A.J.-R.; Writing: I.G.-B., W.H., A.J.-R.; Review: M.K., A.K., A.J.-R. All authors have read and agreed to the published version of the manuscript.

**Funding:** This research received no external funding.

**Conflicts of Interest:** The authors declare no conflict of interest.

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
