# Peer review of "Content and Uptake of Ash and Selected Nutrients (K, Ca, S) with Biomass of Miscanthus × giganteus Depending on Nitrogen Fertilization"

_agriculture, doi:10.3390/agriculture11010076_

Round 1

Reviewer 1 Report

Thanks for addressing my comments. The manuscript looks great. Great job!

There are only a few things that I could suggest fixing before moving forward.

Introduction

  • The first paragraph of the introduction and the first few sentences of the second paragraph are repetitive. It seems both introduce the topic from slightly different angles. I do not believe this is necessary, and may distract the reader. I would suggest combining the information.
  • Previous research on the effects of sulfur and calcium and potassium on biomass combustion is missing. This is a novel component of the manuscript, but the importance of these elements is not presented. How does high S/K/Ca concentration affect combustion? Authors should provide some background information of these effects to guide the reader to why studying these concentrations matters.
    Biomass quality is not my main field of research I just did a quick search on Web of Science and found a few articles that comment on their effects on biomass quality. The paper of  Lewandowski and Kircherer (1997), that you cite, has very nice introduction on the importance of these elements.
    Again, some of this information is in the discussion, it would help the reader to add some to the introduction to show the importance of the topic.

 Methods

  • Authors should still give further detail on how they combined the samples. It great to know that we are finally moving to standardize protocols but we still need to provide detail for someone to repeat the methods.

  • I would also reconsider having the year effect as random. I understand that there is no control over year effects, however, three years should be enough to estimate the variance of years as random effect. Also, if years are considered random, the actual effect at each year should not matter because we only care about the variability, therefore, N fertilization effect shouldn’t be studied by year. I would suggest having years as fixed.

  • Avoid paragraphs of only one sentence.

Results

  • I still think having the organ in the y-axis is not appropriate, I personally have never seen this in other paper. To me is rather misleading. The variable being measured should be the label of the axis.
  • As the data are presented in the graphs two effects are shown (N fertilization and date) but there is only one p-value. Means of the different N treatments are very close so I would think the p-value corresponds to the date effect.

Discussion

  • Looks good, great job!

Author Response

We would like to thank for your insightful comments and the opportunity to submit the manuscript entitled “ Content and uptake of ash and selected nutrients (K, Ca, S) with biomass of Miscanthus x giganteus depending on nitrogen fertilization authorship Izabela Gołąb-Bogacz, Waldemar Helios, Andrzej Kotecki, Marcin Kozak, Anna Jama-Rodzeńska again after corrections. We tried to make an effort to adjust the article to the reviewers' comments.   

Answers on reviewer’s comments:

Reviewer 1:

 Introduction

The first paragraph of the introduction and the first few sentences of the second paragraph are repetitive. It seems both introduce the topic from slightly different angles. I do not believe this is necessary, and may distract the reader. I would suggest combining the information.

Previous research on the effects of sulfur and calcium and potassium on biomass combustion is missing. This is a novel component of the manuscript, but the importance of these elements is not presented. How does high S/K/Ca concentration affect combustion? Authors should provide some background information of these effects to guide the reader to why studying these concentrations matters.
Biomass quality is not my main field of research I just did a quick search on Web of Science and found a few articles that comment on their effects on biomass quality. The paper of  Lewandowski and Kircherer (1997), that you cite, has very nice introduction on the importance of these elements.
Again, some of this information is in the discussion, it would help the reader to add some to the introduction to show the importance of the topic.

We agree with the reviewer that the introduction lacked information on K Ca and S - elements and their importance in the combustion proces.  Introduction has been enriched by information about S/K/Ca role in combustion of the proces (these information has been included in Discussion section and transfered to Introduction with Ca supplementation) as below:

Crude ash content and examined macroelements have a significant impact on the quality of biomass combustion therefore the relevance of these elements is discussed. A high ash concentration decreases the heating value [38,39]. Potassium next to silicon is the main component of ash [12]. The potassium content of biomass is very important because its high content can increase the corrosion effect in heating systems and lower the melting point of ash [38] and is regarded as a critical element in ash-related problems. Therefore the potassium content should be as low as possible [38]. For optimal plant growth, the potassium content should be 10-50 g of DM [38]. Sulphur also play an important role during the combustion process. Sulphur compounds that are formed during this process leading to corrosion and are emitted in to the atmosphere [30]. In turn Calcium can inhibit the occurrence of silicate melt-induced slagging and bed agglomeration, because of forming melting calcium potassium phosphates and silicates at high temperatures.  

Methods

Authors should still give further detail on how they combined the samples. It great to know that we are finally moving to standardize protocols but we still need to provide detail for someone to repeat the methods.

Details on how samples were combined have been added.

Plant samples were sampled from the area of 0.25 m2 by gentle extraction of rhizomes from the soil with the whole stems.

I would also reconsider having the year effect as random. I understand that there is no control over year effects, however, three years should be enough to estimate the variance of years as random effect. Also, if years are considered random, the actual effect at each year should not matter because we only care about the variability, therefore, N fertilization effect shouldn’t be studied by year. I would suggest having years as fixed.

Doses of nitrogen fertilizers were assumed to be fixed factor, while years to be random in our research.  At this stage it would be difficult to change the factors due to the previously published article about Miscanthus with identical methodology, which we refer to in this article.

 Avoid paragraphs of only one sentence.

We will avoid the paragraphs with only one sentence.

I still think having the organ in the y-axis is not appropriate, I personally have never seen this in other paper. To me is rather misleading. The variable being measured should be the label of the axis.

According to your sugetsions we have made a modification of figures, we hope that this change is appropriate. A different way of graphing is not possible in Statistica.

As the data are presented in the graphs two effects are shown (N fertilization and date) but there is only one p-value. Means of the different N treatments are very close so I would think the p-value corresponds to the date effect.

P value presented on the figure concerns the date of plant material sampling. This information has been added to each figures.

We would like to thank for your insightful comments and the opportunity to submit the manuscript entitled “ Content and uptake of ash and selected nutrients (K, Ca, S) with biomass of Miscanthus x giganteus depending on nitrogen fertilization authorship Izabela Gołąb-Bogacz, Waldemar Helios, Andrzej Kotecki, Marcin Kozak, Anna Jama-Rodzeńska again after corrections. We tried to make an effort to adjust the article to the reviewers' comments.   

Answers on reviewer’s comments:

Reviewer 1:

 Introduction

The first paragraph of the introduction and the first few sentences of the second paragraph are repetitive. It seems both introduce the topic from slightly different angles. I do not believe this is necessary, and may distract the reader. I would suggest combining the information.

Previous research on the effects of sulfur and calcium and potassium on biomass combustion is missing. This is a novel component of the manuscript, but the importance of these elements is not presented. How does high S/K/Ca concentration affect combustion? Authors should provide some background information of these effects to guide the reader to why studying these concentrations matters.
Biomass quality is not my main field of research I just did a quick search on Web of Science and found a few articles that comment on their effects on biomass quality. The paper of  Lewandowski and Kircherer (1997), that you cite, has very nice introduction on the importance of these elements.
Again, some of this information is in the discussion, it would help the reader to add some to the introduction to show the importance of the topic.

We agree with the reviewer that the introduction lacked information on K Ca and S - elements and their importance in the combustion proces.  Introduction has been enriched by information about S/K/Ca role in combustion of the proces (these information has been included in Discussion section and transfered to Introduction with Ca supplementation) as below:

Crude ash content and examined macroelements have a significant impact on the quality of biomass combustion therefore the relevance of these elements is discussed. A high ash concentration decreases the heating value [38,39]. Potassium next to silicon is the main component of ash [12]. The potassium content of biomass is very important because its high content can increase the corrosion effect in heating systems and lower the melting point of ash [38] and is regarded as a critical element in ash-related problems. Therefore the potassium content should be as low as possible [38]. For optimal plant growth, the potassium content should be 10-50 g of DM [38]. Sulphur also play an important role during the combustion process. Sulphur compounds that are formed during this process leading to corrosion and are emitted in to the atmosphere [30]. In turn Calcium can inhibit the occurrence of silicate melt-induced slagging and bed agglomeration, because of forming melting calcium potassium phosphates and silicates at high temperatures.  

Methods

Authors should still give further detail on how they combined the samples. It great to know that we are finally moving to standardize protocols but we still need to provide detail for someone to repeat the methods.

Details on how samples were combined have been added.

Plant samples were sampled from the area of 0.25 m2 by gentle extraction of rhizomes from the soil with the whole stems.

I would also reconsider having the year effect as random. I understand that there is no control over year effects, however, three years should be enough to estimate the variance of years as random effect. Also, if years are considered random, the actual effect at each year should not matter because we only care about the variability, therefore, N fertilization effect shouldn’t be studied by year. I would suggest having years as fixed.

Doses of nitrogen fertilizers were assumed to be fixed factor, while years to be random in our research.  At this stage it would be difficult to change the factors due to the previously published article about Miscanthus with identical methodology, which we refer to in this article.

 Avoid paragraphs of only one sentence.

We will avoid the paragraphs with only one sentence.

I still think having the organ in the y-axis is not appropriate, I personally have never seen this in other paper. To me is rather misleading. The variable being measured should be the label of the axis.

According to your sugetsions we have made a modification of figures, we hope that this change is appropriate. A different way of graphing is not possible in Statistica.

As the data are presented in the graphs two effects are shown (N fertilization and date) but there is only one p-value. Means of the different N treatments are very close so I would think the p-value corresponds to the date effect.

P value presented on the figure concerns the date of plant material sampling. This information has been added to each figures.

We would like to thank for your insightful comments and the opportunity to submit the manuscript entitled “ Content and uptake of ash and selected nutrients (K, Ca, S) with biomass of Miscanthus x giganteus depending on nitrogen fertilization authorship Izabela Gołąb-Bogacz, Waldemar Helios, Andrzej Kotecki, Marcin Kozak, Anna Jama-Rodzeńska again after corrections. We tried to make an effort to adjust the article to the reviewers' comments.   

Answers on reviewer’s comments:

Reviewer 1:

 Introduction

The first paragraph of the introduction and the first few sentences of the second paragraph are repetitive. It seems both introduce the topic from slightly different angles. I do not believe this is necessary, and may distract the reader. I would suggest combining the information.

Previous research on the effects of sulfur and calcium and potassium on biomass combustion is missing. This is a novel component of the manuscript, but the importance of these elements is not presented. How does high S/K/Ca concentration affect combustion? Authors should provide some background information of these effects to guide the reader to why studying these concentrations matters.
Biomass quality is not my main field of research I just did a quick search on Web of Science and found a few articles that comment on their effects on biomass quality. The paper of  Lewandowski and Kircherer (1997), that you cite, has very nice introduction on the importance of these elements.
Again, some of this information is in the discussion, it would help the reader to add some to the introduction to show the importance of the topic.

We agree with the reviewer that the introduction lacked information on K Ca and S - elements and their importance in the combustion proces.  Introduction has been enriched by information about S/K/Ca role in combustion of the proces (these information has been included in Discussion section and transfered to Introduction with Ca supplementation) as below:

Crude ash content and examined macroelements have a significant impact on the quality of biomass combustion therefore the relevance of these elements is discussed. A high ash concentration decreases the heating value [38,39]. Potassium next to silicon is the main component of ash [12]. The potassium content of biomass is very important because its high content can increase the corrosion effect in heating systems and lower the melting point of ash [38] and is regarded as a critical element in ash-related problems. Therefore the potassium content should be as low as possible [38]. For optimal plant growth, the potassium content should be 10-50 g of DM [38]. Sulphur also play an important role during the combustion process. Sulphur compounds that are formed during this process leading to corrosion and are emitted in to the atmosphere [30]. In turn Calcium can inhibit the occurrence of silicate melt-induced slagging and bed agglomeration, because of forming melting calcium potassium phosphates and silicates at high temperatures.  

Methods

Authors should still give further detail on how they combined the samples. It great to know that we are finally moving to standardize protocols but we still need to provide detail for someone to repeat the methods.

Details on how samples were combined have been added.

Plant samples were sampled from the area of 0.25 m2 by gentle extraction of rhizomes from the soil with the whole stems.

I would also reconsider having the year effect as random. I understand that there is no control over year effects, however, three years should be enough to estimate the variance of years as random effect. Also, if years are considered random, the actual effect at each year should not matter because we only care about the variability, therefore, N fertilization effect shouldn’t be studied by year. I would suggest having years as fixed.

Doses of nitrogen fertilizers were assumed to be fixed factor, while years to be random in our research.  At this stage it would be difficult to change the factors due to the previously published article about Miscanthus with identical methodology, which we refer to in this article.

 Avoid paragraphs of only one sentence.

We will avoid the paragraphs with only one sentence.

I still think having the organ in the y-axis is not appropriate, I personally have never seen this in other paper. To me is rather misleading. The variable being measured should be the label of the axis.

According to your sugetsions we have made a modification of figures, we hope that this change is appropriate. A different way of graphing is not possible in Statistica.

As the data are presented in the graphs two effects are shown (N fertilization and date) but there is only one p-value. Means of the different N treatments are very close so I would think the p-value corresponds to the date effect.

P value presented on the figure concerns the date of plant material sampling. This information has been added to each figures.

Reviewer 2 Report

Dear Authors,

I have studied your resubmission version of your article. I am of the opinion that the quality of the article has significantly increased. Nevertheless, I have a few comments, especially on statistical data processing.

For the sake of clarity, I have processed all my comments into a pdf document.

Author Response

Reviever 2

2.1 Study site and fertilisation treatments of Miscanthus x giganteus: Line 125 … (Photo 1) …. I recommend using the Figure. Line 211 … Photo 1. Location of experiment … I recommend adding the coordinates of the field experiment, the scale and the north to the description of the map.

We changed the photo of location and the expression as reviewer suggested.

The data are well described, I appreciate that, on the other hand, I can not completely verify your conclusions, because the graphs and tables are not clear. As was the case last article review, I must say that the data are well documented, but the problem I see in the representation of significance of shown differences.

From Tables 1-8 is not evident whether there is a significant difference between the individual years of measurement and the variants. I suggest again to show significant differences in graphs and tables, or to add comments under pictures and graphs. Because I'm not sure if we understand each other, I decided on a graphic representation. Perhaps it will help explain the problem:

According to the suggestions we add significant differences in Tables as a Homogeneous groups that were determined on the basis of the Tukey test (included in statistical analysis).

In the figures only one P value is added (conserns the date of plant sampling). Ewentually we can change these figures into table, perhaps with other significant differences presenting and attached as a suplemmentary file, however this changes could not be clear and readable.

This manuscript is a resubmission of an earlier submission. The following is a list of the peer review reports and author responses from that submission.

Round 1

Reviewer 1 Report

The article presents results on the effects of nitrogen fertilization on leaf, stem and rhizome K, Ca, S and ash content. I commend the authors for such an intensive field research done over three years. The temporal resolution seems very appropriate for the study. I also enjoyed they considered leaves, stems, and also below-ground organs. I am also sure that the research would be of interest to the readers of Agronomy Journal, however, the manuscript is still in the early stages. The manuscripts presents relevant pitfalls that must be addressed.

  • The introduction does not do a great job at introducing the relevance of the knowledge gap. I would encourage the authors to support the sentences with data to create an idea of the relevance of the problem. Also, I think is crucial to state what are the potential problems that Ca, K and S may cause during combustion and why they deserved to be studied.
    I also want to point out that the introduction is centered around the combustion, however, a lot of the results are about the internal dynamic of these elements. I found it misleading as a reader.
  • The hypothesis should be refined to testable statements. What would be the direction of the change? Would N fertilization increase their concentration? decrease it?. 
  • The material and methods sections need much more attention. Authors have to provide better description of the analyses and also how samples were collected.
    The experimental design seems sounds but the brief description of some the methods makes the research hard to replicate.
  • The results were a bit too descriptive of the data and may not guide the reader to the specific points that will support (or not) the hypotheses, and will then be discussed. Also, variables should be in the y-axis, not the organ!
  • The discussion does a good job a compiling some information, maybe some of it could go in the introduction. However, at times it fails to guide the reader to a specific point and ends up summarizing previous research and not discussing their results.

Please, see the attached pdf for inline comments and further detail on the above-mentioned issues.

Cheers,

Reviewer 2 Report

Dear Authors,

I have studied your article. However, it will be necessary to make some changes and edit the article. Because in its current form it contains some inaccuracies and potential mistakes. The presented topic is very current and important, but for admission to a scientific journal, the manuscript should be improved.

Reviewer

Abstract

I recommend to add some results - one or two sentences. I lack information on how the addition of nitrogen affected the uptake of nutrients. For example: calcium uptake by the plant increased by 10% (significantly), etc.

Introduction

This part is brief and clear - general aspects of the topic are described. I suggest adding a hypothesis at the end of this chapter, which could be used to describe the result and discuss them.

Materials and Methods

2.1 Study site and fertilisation treatments of Miscanthus x giganteus:

I have problem with the information about geographic location of your field/terrain experiment in first paragraph.

Line 87 .... treatments are described in Bogacz et al. 2020

I assume that your work follows in a way the experiment "Bogacz et al. (2020)". But you should still provide an overview of the experiment variations and, for example, a location map. I also see the problem in the fact that only in the results chapter will the reader learn more about the organization of the experiment.

2.2. Chemical analysis of plant material

Line 96 …. Samples for chemical analysis were prepared according to the standard requirements of PN-EN ISO 14780:23017-07 …

  • I assume that it was only the collection of plant biomass (It is not described in the text). Above all, I lack information on how many samples, from which area and from which variants it was taken.

2.3. Statistical analysis

Line 105 …. The experiment was conducted with a randomised block design in four replications …

  • This information should be given earlier in the description of the field experiment, including an overview of variants and their treatment.

Line 107 …. The analysis of variance (ANOVA) and a mixed model with repeated measurements were …

  • It should be stated whether one-way or two-way analysis of variance was used. Furthermore, the level of significance (? P < 0.01. or 0.05 ?) at which the analyzes were performed is not stated. ANOVA alone cannot determine the significance of differences in the measured data, it is necessary to perform a post-hoc analysis (for example Tukey´s HSD test or Fischer LSD test etc). Was performed?

Results

3.1. Crude ash content and uptake 3.4 Sulphur content and uptake

The data are well described, I appreciate that, on the other hand, I can not completely verify your conclusions, because the graphs and tables are not clear. For example:

Table 1 Crude ash content in dry matter …..

  • The table does not contain information on how many values (n = 4 ??) the differences were calculated for individuals variant, nor are the values of the standard error. Does the table show average values? I recommend adding “letters” to show the evidence of the differences (P <0.05 or 0.01) between variants "0" and "60". Furthermore, I assume that the table shows in the first part the average for 3 years and in the second part for individual years (I assume a different size of the statistical data set). But it is not so obvious from the table without a doubt.

Figure 1 Crude ash content in examined part ….

  • The graphs do not contain any information on the presence or absence of statistically significant differences. This applies to all of the following graphs (Figure 1 – Figure 8). Without adding this data to the graph, it is not clear whether what is written in the text is true.
  • I assume that these are average values in individual months of sampling for three years of measurement. However, this is not clear from the description of the individual Figures.
  • If a difference was found between the individual months or variants, it would be appropriate to state on what basis. For example, a paired t-test?

I suggest to show significant differences in graphs and tables, or to add comments under pictures and graphs.

Discussion

Line 223 - Nutrient remobilization seems to be a good strategy for perennial rhizomatous grasses and represents an environmentally friendly strategy to reduce fertiliser applications …

  • This is a very good idea, I recommend expanding it more. This could be one of the main benefits.

Line 267 - Their experiment also indicated that many genotypes of Miscanthus x giganteus were characterized by a higher concentration of potassium in autumn …

  • This is a very good idea, I recommend expanding it more. This could be one of the main benefits.

Line 275 - The uptake of macronutrients is strongly dependent on the yield …

  • From a general point of view, you are right, but I'm not sure if you can make this statement based on the data in the article. Because the two main variants were 0 and 60 kg N, not variants with different genotypes.

Line 301 - In our research, the trends of changes in calcium content during vegetation were similar to those …

  • It is possible, but from the point of view of data interpretation it cannot be confirmed. Measured trends are not statistically processed in your work. It can of course be supplemented by the analysis of time series, ie by comparing the average values of nutrient uptake in individual months between variants and then for the whole period

Conclusion

Line 324 - While the research hypothesis was verified …

  • Your work contains a goal, I did not find a hypothesis. As I pointed out above, I recommend adding it at the end of the introduction chapter